# Genome-Wide Investigation of *BAM* Gene Family in *Annona atemoya*: Evolution and Expression Network Profiles during Fruit Ripening

**DOI:** 10.3390/ijms241310516

**Published:** 2023-06-22

**Authors:** Luli Wang, Minmin Jing, Shuailei Gu, Dongliang Li, Xiaohong Dai, Zhihui Chen, Jingjing Chen

**Affiliations:** 1Key Laboratory of Tropical Fruit Biology, Ministry of Agriculture and Rural Affairs, South Subtropical Crops Research Institute, Chinese Academy of Tropical Agricultural Sciences, Zhanjiang 524091, China; luli_wang@catas.cn (L.W.); jingminmin@catas.cn (M.J.); gushuailei@catas.cn (S.G.); lidongliang@catas.cn (D.L.); daixiaohong@catas.cn (X.D.); czhihui168@163.com (Z.C.); 2Key Laboratory of Hainan Province for Postharvest Physiology and Technology of Tropical Horticultural Products, South Subtropical Crops Research Institute, Chinese Academy of Tropical Agricultural Sciences, Zhanjiang 524091, China

**Keywords:** fruit ripening, β-amylase proteins, *Annona atemoya*

## Abstract

β-amylase proteins (BAM) are important to many aspects of physiological process such as starch degradation. However, little information was available about the *BAM* genes in *Annona atemoya*, an important tropical fruit. Seven *BAM* genes containing the conservative domain of glycoside hydrolase family 14 (PF01373) were identified with *Annona atemoya* genome, and these *BAM* genes can be divided into four groups. Subcellular localization analysis revealed that *AaBAM3* and *AaBAM9* were located in the chloroplast, and *AaBAM1.2* was located in the cell membrane and the chloroplast. The *AaBAMs* belonging to Subfamily I contribute to starch degradation have the higher expression than those belonging to Subfamily II. The analysis of the expression showed that *AaBAM3* may function in the whole fruit ripening process, and *AaBAM1.2* may be important to starch degradation in other organs. Temperature and ethylene affect the expression of major *AaBAM* genes in Subfamily I during fruit ripening. These expressions and subcellular localization results indicating β-amylase play an important role in starch degradation.

## 1. Introduction

Amylases catalyze starch and convert it into sugars such as glucose and maltose. α-1,4 glycosidic linkages in starch are hydrolyzed by β-amylase to remove successive maltose units from the non-reducing ends of the chains [1,2]. β-amylases are divided into *GH-14B* presenting in bacteria and *GH-14A* found in plants according to the sequence similarity between the glycoside hydrolase 14 family (*GH-14*) [3,4,5]. β-amylase structure is composed of a canonical (βα) eight-barrel core and a C-terminal long loop, and the active site is located in the deep pocket of the (βα) eight-barrel core [6].

The analysis of the conservation of intron positions in the *BAM* family in land plants revealed that there are two subfamilies of *BAMs* [7]. Moreover, the amino acid sequence alignments suggested that these proteins are divided into four distinct clades [8]. Subfamily I contains *Arabidopsis BAM1*, *BAM3* and *BAM9*, and Subfamily II contains *BAM2* and *BAM4* through *BAM8*. From the evolution trend of plant *BAM* gene, it can be inferred that *BAM1/3* and *BAM2* are probably the ancestral forms of each subfamily in land plants. Other *BAM* genes in *Arabidopsis* and angiosperms probably produced by duplication and evolution change from these two original *BAMs* [9].

*BAM* genes are widely distributed among the plant kingdom: 9 in *Arabidopsis*, 10 in rice, 13 in maize, 15 in banana, 10 in *Sorghum* [10]. It is known that β-amylase is the main way for starch degradation of *Arabidopsis* leaves, potatoes tubers, fruits, etc. [8,11,12,13,14,15,16]. There are many studies on nine *AtBAM* genes in *Arabidopsis*, but the function of those genes is not well demonstrated. *BAM1* to *BAM4*, *BAM6*, and *BAM9* are located in the plastid, *BAM7* and *BAM8* are located in the nucleus, and *BAM5* is located in the cytoplasm. Only *AtBAM1* and *AtBAM3* are directly involved in starch degradation, playing a central role in the starch decomposition of leaves [8,9,13,16,17,18]. *BAM1*, *BAM3*, *BAM5* and *BAM6* of *Arabidopsis* have catalytic activity in addition to *BAM4*, *BAM7*, *BAM8* and *BAM9* [9].

*Annona atemoya* is an economically valuable fruit crop cultivated. *Annona atemoya* belongs to the family of *Annonaceae Annona*, also known as Sakya, Matri, and Fotou, native to tropical America. It is a semi-deciduous, exotic subtropical fruit that is consumed in various countries [19]. At present, *Annona squamosa* Linn and *Annona atemoya* Hort are the main cultivated varieties. They are mainly planted in southeast coastal areas in mainland China. The *Annona atemoya* planting area in China reached 11,889.15 hectares in 2021, displaying a growing trend in recent years. These plants are popular because of their fast growth, early bearing, stability to yield, good fruit quality, aromatic flavor, and high edible and medicinal value. But the molecular and genetic mechanisms of fruit development have not been explored extensively. The main form of carbohydrate storage in *Annona atemoya* is starch, accounting for 10–12% of the fresh weight [20]. Starch, as the content of fruit cells, supports the cell wall. The transformation of starch to soluble sugar significantly affects the expansion of fruit cells, and then participates in fruit softening by affecting the tension in fruit cells [21,22].

Owing to the significant role of starch degradation in the process of fruit ripening and softening after harvest, we intended to explore the impact of *AaBAM* genes on fruit ripening. We then conducted a series of studies on genome-wide level. The β-amylase gene family was screened and cloned, and its gene structure, phylogeny, and subcellular localization were analyzed. Subsequently, the tissue specificity, expression patterns under different temperatures, ethylene and auxin treatments were emphatically researched to determine the function of *AaBAMs*. This research provided an insight into the important role of β-amylase genes in *Annona atemoya* to starch degradation and fruit ripening, and a theoretical basis of the preservation of *Annona atemoya* after harvest.

## 2. Results

### 2.1. Identification and Classification of AaBAMs Genes in Annona atemoya

Initially, nine *Annona atemoya BAM* genes were identified by the Hidden Markov Model (HMM) search. The annotation of these gene models was further checked using transcriptome data. Seven redundant predicted *BAM* genes were manually curated and two *AaBAM8* redundant sequences were then removed. Finally, seven gene models were selected and annotated as *BAM* genes based on the presence of apparently complete BAM domains (Appendix A). All the *BAM* genes in *Annona atemoya* could be positioned on seven linkage groups, respectively.

The length of the CDS (Coding Sequence), length of the protein sequence, protein molecular weight (MW), Isoelectric point (pI), and subcellular localization are shown in Appendix A. *AaBAM9* was identified to be the smallest protein with 536 amino acids (aa), whereas the largest one was *AaBAM5* (899 aa), and the length of four of seven genes 536–588 amino acids (aa). The molecular weight of the corresponding coding protein is 59.23–101.9 kDa, and protein IP ranged from 5.70 to 8.39.

### 2.2. Multiple Sequence Alignment, Phylogenetic Analysis, and Classification of AaBAM Genes

To study the evolutionary relationship between *AaBAM* genes and the known *BAM* genes from *Arabidopsis* and *Oryza sativa*, multiple sequence alignments were conducted and then a phylogenetic tree was constructed based on amino acids of *BAM* genes in *Arabidopsis*, *Glycine max*, *Citrus trifoliata*, and rice. These results indicated the *BAM* genes can be divided into four subfamilies: Group I including *AaBAM1* and *AaBAM3*, Group II including *AaBAM5* and *AaBAM6*, Group III including *AaBAM2*, *AaBAM7* and *AaBAM8*, Group IV including *AaBAM4* and *AaBAM9* (Figure 1). *BAMs* of rice and trifoliate orange are distributed in four subfamilies, and seven *AaBAM* genes are distributed in four subfamilies with one in Group II and IV, two in Group III, and three in Group I (Figure 1). Compared with rice, *Annona atemoya* is closely related to *Arabidopsis thaliana*.

Alignment of multiple *AaBAM* was conducted to gain more insight into the structure and function of the BAM family in *Annona atemoya*. β-amylase structure is composed of canonical (βα) eight barrel cores and a C-terminal long loop, and the structures of β-amylases are known and show that these proteins contain a TIM-barrel fold with a pair of catalytic glutamates in the active site that is required for cleavage of the glycosidic bond [9]. All *AaBAM* genes identified in *Annona atemoya* were conserved in catalytic site E380 of soybean *GmBAM5* (Appendix A). For catalytic site E186 of *GmBAM5*, only *AaBAM9* was not conserved and had an amino acid of *Glycine* other than glutamate (Appendix A). All of the BAM proteins aligned well in the core amylase domain, with most of the differences occurring to the N and C-terminal sides of this domain. Conformational change in some residues close to the active site was induced by sugar binding. Movement of the loop composed of a conserved GGNVGD sequence was observed among *AaBAM3*, *AaBAM5*, *AaBAM1.2*, and *AaBAM7*.

### 2.3. Gene Structure and Conserved Motif Analysis

To determine the more evolution pattern of *AaBAM* genes, exon–intron organizations of all the identified genes were examined. The result showed that the closely related genes were usually more similar in gene structure. For instance, genes *AaBAM7* and *AaBAM8* belonging to Group III have nine exons, and *AaBAM3* and *AaBAM1.1* belonging to Group I have four exons. However, some closely related genes showed some difference in structural arrangements; for example, *AaBAM1.1* has four exons while *AaBAM1.2* has five exons (Figure 2). Furthermore, *AaBAM* genes from four groups showed distinct numbers of exons with 9 in Group III, 13 in Group II, 3 in Group IV, 4–5 in Group I, respectively (Figure 2).

Twenty conserved motifs were identified in the AaBAM proteins by MEME programs. As exhibited in Figure 2, AaBAM members within the same groups were usually found to share a similar motif composition other than Motifs 1, 3, 6, 4, 8, 5, 2, 7 and 9 which are widely distributed (Appendix A). Moreover, the common motifs existed in all AaBAM genes that were arranged in the same order. To character the motif pattern in a particular BAM group, the rest of the motifs were calculated. For example, Motif 10 and Motif 12 are uniquely present in AaBAM7 and AaBAM8 belonging to Group III. From the perspective of motif amounts in BAM groups, AaBAM1.1 and AaBAM1.2 in Group I have the same motif composition, and they have nine motifs. AaBAM9 in Group IV has 13 motifs, and AaBAMs belonging to Group II have the highest number of motifs which is 17. These result shows that the *AaBAM* genes of different groups have distinct patterns, and the motif compositions or gene structures of the *BAM* members in the same group are similar. Together with the phylogenetic analysis results, this evidence could strongly support the reliability of the group classifications.

### 2.4. The Location and Synteny Relationship Analysis of BAM Genes

A total of seven genes were identified in five Chromosomes, while the Chr03 and Chr06 had no *BAM* genes (Figure 3). The number of *BAM* genes on one chromosome was between one and two. chromosome Chr01, Chr04, Chr05, and Chr07 each had one *BAM* gene, and Chr02 contained the largest number of two *BAM* genes (Figure 3). There was a positive correlation between the chromosomes’ length and the number of *BAM* genes.

### 2.5. Expression Analysis of the AaBAM Genes with RNA-seq

The expression patterns of all seven *AaBAM* genes in the transcriptome data, which was derived from different developmental stages of *Annona atemoya* organs/tissues including young leave, root, flower bud, twig, young fruit and seed, were investigated in this research (Figure 4). All seven *AaBAM* genes were detected to express at least one tissue, with *AaBAM3*, *AaBAM1.2*, *AaBAM7*, *AaBAM1.1*, *AaBAM8*, *AaBAM9* existing in six organs, and *AaBAM5* only expressed in young fruit. The results of the expression pattern reveal that *AaBAM1.2* had the highest expression level among the *AaBAM* genes and the highest value of expression in the tissue of young leave and root. In addition, *AaBAM1.1*, *AaBAM3* expressed lower than other genes in the same family, but had a higher expression in root and twig. Except *AaBAM1.2*, the rest of the *AaBAM* genes were evenly expressed in those tissues. *AaBAM8* and *AaBAM9* had a relatively high expression in seed and root, respectively.

*AaBAM* genes were randomly selected for quantitative RT-PCR analysis of young fruit and seed to verify the RNA-seq data (Figure 4). The qRT-PCR results confirmed that almost all *AaBAM* genes had expression in most of the organs, and most of the *AaBAM* genes expressed higher in seed than in young fruits. These results showed that our RNA-seq data are suitable for investigating the expression patterns of *AaBAM* genes in different tissues.

### 2.6. Expression Patterns of Annona atemoya Genes in Response to Temperature Treaments

The *Annona atemoya* fruits were treated with 32 °C, 28 °C and 15 °C in this study. The analysis of the expression showed that *AaBAM3* was the highest in the whole fruit ripening process (Figure 5C). The expression of *AaBAM3* genes showed an overall upward trend under the three temperature treatments, indicating that temperature may have an effect (Figure 5C). Further evidence shows that a 15 °C treatment inhibited the expression of *AaBAM3* genes, consistent with the result of delaying maturity at 15 °C (Figure 5C).

The expression of *AaBAM9* was up-regulated in the first 4 days of the three temperature treatments. However, after the fourth day, the expression of *AaBAM9* decreased at 15 °C and 32 °C, while the expression continued to increase at 28 °C (Figure 5E). The expression trends at 28 °C and 32 °C were consistent with *AaBAM3*, indicating that both *AaBAMs* may have the same response expression in these two temperature conditions (Figure 5E).

Both *AaBAM1.1* and *AaBAM1.2* showed an expression trend of first rising and then declining under 32 °C and 28 °C treatments (Figure 5A,B). After 15 °C treatments, they showed an overall trend of inhibiting expression (Figure 5A,B). Finally, the overall expression of *AaBAM7* and *AaBAM8* was low (Figure 5D,F).

On the whole, *AaBAM3*, *AaBAM9*, *AaBAM1.1* and *AaBAM1.2* with high expression levels were up-regulated under 28 °C treatments, especially for the first 4 days, indicating that 28 °C treatments could accelerate starch degradation (Figure 5). At 15 °C, the overall expression of *AaBAM3*, *AaBAM9*, *AaBAM1.1* and *AaBAM1.2* was lower than that at 28 °C and 32 °C, which was consistent with the results of delaying starch degradation at low temperature (Figure 5).

### 2.7. Expression Patterns of Annona atemoya Genes in Response to Ethylene and Auxin Treatments

The expression of *AaBAM3* began to increase significantly after 24 h of ethylene treatment, while IAA treatment induced the expression of *AaBAM3* between 1 and 2 days (Figure 6). The expression of *AaBAM9* was up-regulated under ethylene and IAA treatment, and the induction effect of ethylene was the most obvious (Figure 6E). The expression of *AaBAM9* was induced by ethylene and auxin during the first 24 h (Figure 6E). *AaBAM8* showed induced expression trends under IAA treatment, while the overall expression changed slightly (Figure 6D). Under ethylene and IAA treatment, the expression of *AaBAM1.1* and *AaBAM1.2* was irregular before 12 h, and was induced after 12 h (Figure 6A,B). The results showed that ethylene significantly induced the expression of *AaBAM3* and *AaBAM9* genes, and IAA induced the expression of *AaBAM8* and *AaBAM9* (Figure 6E).

### 2.8. Subcellular Localization Analysis of AaBAM Protiens

To study the subcellular localization of *AaBAM* proteins, we generated BAM–GFP fusion constructs and transiently expressed them in *Nicotiana benthamiana* (tobacco) leaves (Pro35S:BAM-GFP). In agreement with the presence of a predicted chloroplast localization signal, each BAM protein showed exclusively chloroplast localization (Figure 7). The results showed that the empty vector with GFP was expressed in the membrane and nucleus of tobacco leaf cells. *AaBAM1.2-GFP* was only expressed in the membrane and chloroplast, but not in the nucleus (Figure 7). *AaBAM3-GFP* was expressed in chloroplasts, which shows that the fluorescence channels of chloroplasts overlap with the *AaBAM3-GFP* protein channels (Figure 7). *AaBAM9-GFP* is similar to *AaBAM3-GFP*, and it is also located in chloroplast according to the analysis of subcellular localization (Figure 7).

## 3. Discussion

The analysis of *BAM* gene families was carried out in some special in *Arabidopsis*, rice, maize, banana, and *Sorghum*, respectively [10]. In this study, a search for *BAM* gens in the *Annona atemoya* genome resulted in the identification of seven members, which were designated on the basis of homology analysis. Amino acid sequence alignments indicate that these proteins fall into four distinct clades, and that none of these proteins are more than 60% identical. This revealed that none of the genes of *AaBAM* resulted from very recent gene duplications.

Previous analysis of the *BAM* family in land plants indicated that there are two subfamilies of *BAMs* that became separated prior to the origin of land plants [7]. Subfamily I contains *Arabidopsis BAM1*, *BAM3* and *BAM9*, whereas Subfamily II contains *BAM2* and *BAM4* through *BAM8*. *BAM1/3* and *BAM2* are proposed to be the original forms of each subfamily of land plants. Unlike vascular plants that contain both *BAM1* and *BAM3* genes, *BAM1* and *BAM3* are not easily distinguished from each other in *brophytes Physcomitrella Patens* and *Marchantia polymorpha*, so they are defined as *BAM1/3* [9]. In *Annona atemoya*, *AaBAM1.1* cannot be defined as *BAM1* or *BAM3* according to homology, so *AaBAM1.1* was identified as *BAM1/3*. *Annona atemoya* contained all types of Subfamily I *BAM* genes, while the only existing Subfamily II *BAM* genes were *AaBAM5*, *AaBAM7* and *AaBAM8*.

BAM plays an important role in starch degradation and serves as a major enzyme in hydrolytic process of linear glucans in the plastid. The degraded maltose is exported to the cytosol by a specific chloroplast membrane protein, which is a maltose transporter, to maintain the cycle balance of sucrose–starch metabolism [23,24]. At present, there are many studies on the nine *AtBAM* genes of *Arabidopsis*, but not all the functions of each family member have been clarified. In *Arabidopsis*, *AtBAM1* and *AtBAM3,* which belong to Type I *BAM* genes, are critical for plant starch degradation and function in the process of linear glucans in the plastid [8,13,17,25,26]. The subcellular localization of *AaBAM3*, *AaBAM9* and *AaBAM1.2* found that Type I *AaBAM* genes were located in the chloroplast. Combined with the expression pattern, these results indicated the Type I *BAM* genes in *Annona atemoya* may serve the conserving function as *Arabidopsis*. For *AaBAM9* as Type I genes, the amino acid of catalytic site as glutamate was replaced by glycine and the expression pattern consistent with *AaBAM3* in temperature, auxin, and ethylene treatment, indicating that *AaBAM9* perhaps lost the function of the hydrolytic process but influenced the metabolism balance as a regulated role.

Knowing where the genes are expressed is important for understanding the molecular mechanisms of biological process. It was recently discovered that the BAM protein has a remarkably short half-life, such as, for example, *BAM3* (0.43 days), indicating that control of the rate of *BAM3* expression is important [27]. The extensive expression of the *AaBAM* gene in *Annona atemoya* indicates that degradation of starch into monosaccharide occurs widely in various tissues, and *AaBAM1.2* may play a major role in this process owing to its high expression in most kinds of tissue under the development state of a plant. A lot of evidence demonstrates that *BAM3* plays a prominent role in nocturnal starch degradation in chloroplasts and is expressed in mesophyll cells [8,17,28]. The expression of *AaBAM3* in leaves is relatively high, suggesting *AaBAM3* and *AaBAM1* may jointly regulate starch degradation in leaves. *AaBAM5* is uniquely expressed in young fruit, indicating it may have some function in fruit development. *BAM5* was found to be elevated in a series of starchless *Arabidopsis* mutants and its transcriptional function to be induced by sugars, especially sucrose [29,30,31]. These results of *AaBAM5* analysis suggest this gene may participate in the degradation of starch together with the *AaBAM* gene family, especially during fruit ripening. The lacking expression of *AaBAM5* indicates that starch degradation in organs other than young fruit is mainly accomplished by other β-amylases.

The transformation of starch into soluble sugar significantly affects the pressure on fruit cells, and then participates in fruit softening by affecting the tension in fruit cells. Previous research shows that the β-amylase gene can be regulated by ethylene, auxin, sugar, temperature and light. Furthermore, β-amylase genes can be induced and regulated by cold in *Arabidopsis* leaves and potato tubers [13,17,31,32,33,34,35]. These results reveal that these factors may mediate the ripening process of *Annona atemoya* by regulating the *AaBAM* genes family. There are two genes, *AaBAM1.1* and *AaBAM1.2*, which are annotated as *BAM1* in *Annona atemoya*. The two genes also have a high level of expression during fruit ripening, but lower than *AaBAM3*. Combined with the tissue-specific expression analysis, it is believed that *AaBAM1.1* and *AaBAM1.2* belong to the universal expression genes and are involved in starch degradation of fruit. However, *AaBAM3* probably is the main gene that functions in degrading starch and fruit ripening with its dramatic increase in expression after harvest from tree. Its expression pattern is consistent with the fruit ripening process under the three-temperature condition (Figure 5C). Contrary to the research results asserting that *BAM* gene expression of *Arabidopsis* leaves and potato tubers is induced by low temperature [13,17,34,35], low-temperature treatment in fruit reduces the expression of major β-amylase genes. Compared with normal temperature, the activity of β-amylase is inhibited and the degradation of starch is delayed. This suggests that the regulation mechanism of starch degradation in response to low temperature in fruit may be different from that in leaves, tubers and other tissues or organs.

The role of ethylene in fruits has been widely studied. It has been shown that ethylene can accelerate fruit ripening and softening [36]. 1-Aminocyclopropylene-1-carboxylic acid (ACC) and IAA enhance the softening and electrical conductivity of the treated fruit, and the respiration of fruits treated with IAA is enhanced. In this study, The expression of most of the *AaBAMs* genes in the fruit were induced by IAA and ethylene to compare with the control, including *AaBAM1.2*, *AaBAM3*, *AaBAM9*, *AaBAM7*, and *AaBAM8*. The most significant expression change is that of *AaBAM3* under ethylene treatment, further suggesting *AaBAM3* may play an important role in fruit ripening and softening (Figure 6C). Ethylene treatment significantly induced the expression of *AaBAM3* and *AaBAM9* genes, also suggesting that starch degradation is related to the process of fruit ripening. IAA induced the expression of *AaBAM8* and *AaBAM9*, indicating the *AaBAM7* and *AaBAM8* may be regulated by the auxin signal (Figure 6E).

Compared with other *BAM* genes, the expression level of *AaBAM7* and *AaBAM8* was lower under each treatment. Studies in *Arabidopsis* determined that *AtBAM7* and *AtBAM8* are located in the nucleus and have no catalytic activity, but potential homologous to the BES1/BZR1 domains were identified in transcription factors that respond to brassinosteroid (BR) signaling [8,9,37,38]. In *Annona atemoya*, whether *AaBAM7* and *AaBAM8* have catalytic or regulation functions needs continued study in the future.

In conclusion, a comprehensive analysis of the BAM gene family in *Annona atemoya* was carried out in the present study. Phylogenetic comparison of *BAM* genes from several different plant species provided valuable clues about the evolutionary characteristics of *Annona atemoya BAM* genes. Seven *BAM* genes were classified into four main groups, with high similar exon–intron structures and motif compositions within the same groups. *AaBAM* genes are significant to *Annona atemoya* growth, and mature process as indicated by their expression patterns of different tissues and under physical and physiological treatments. Combined with subcellular localization analysis results, it can be shown that the *AaBAMs* belonging to Subfamily I contribute to starch degradation, especially for relatively high-expression *AaBAM* genes such as *AaBAM1.1*, *AaBAM1.2*, *AaBAM3*. These analyses lay a foundation for the functional analysis of *AaBAM* genes and provide a valuable resource for better understanding the biological roles of individual *BAM* genes in the *Annona atemoya*.

## 4. Materials and Methods

### 4.1. Gene Identification

The Hidden Markov Model (HMM) file corresponding to the BAM domain (PF01373) was downloaded from the Pfam protein family database (http://pfam.sanger.ac.uk/ (accessed on 1 September 2022)). HMMER was used to search the *BAM* genes from the *Annona atemoya* genome database by default parameters. The potential genes were then manually examined to ensure the conserved sequence of the predicted BAM domain.

RNA-seq data were used to further check the annotation of the predicted *BAM* gene models [39]. The incorrectly predicted genes were then manually adjusted and were validated by PCR. The redundant sequences were discarded. The length of sequences, molecular weights, isoelectric points and subcellular location predication of identified *AaBAM* proteins were obtained by using tools from the ExPasy website (https://web.expasy.org/cgi-bin/protparam/protparam/ (accessed on 5 September 2022)), ProtComp9.0 website and Plant-mPLoc Server (http://www.csbio.sjtu.edu.cn/bioinf/plant-multi/ (accessed on 12 September 2022)).

### 4.2. Classification of AaBAM Genes

*BAM* genes in *Arabidopsis*, *Glycine max*, *Citrus trifoliata*, and rice were used for classifying the *BAM* genes. The BAM domain sequences of the characterized BAM proteins were used to create multiple protein sequence alignments using ClustalW with default parameters, and the phylogenetic tree was further annotated by the iTOL program (http://itol.embl.de/ (accessed on 15 October 2022)).

### 4.3. Gene Distribution, Structure, and Motif Analysis

The MEME online program (http://meme.nbcr.net/meme/intro.html/ (accessed on 27 October 2022)) for protein sequence analysis was used to identify conserved motifs in the identified *AaBAM* proteins [40]. The optimized parameters were employed as follows: the number of repetitions is any; the maximum number of motifs is 20; the optimum motif width is set between 6 and 200. *AaBAM* genes identified in this study were mapped to *Annona atemoya* chromosome base on physical location information of genome by the mapchart program.

### 4.4. Location of AaBAM Genes on Chromosomes

To explore the chromosomal location of *AaBAM* genes, the software (https://www.wur.nl/en/show/Mapchart.htm/ (accessed on 1 December 2022)) was used to map *AaBAM* genes onto chromosomes according to the *AaBAM* gene position on the genome of *Annona atemoya*.

### 4.5. Expression Analysis of Annona atemoya BAM Genes in Four Tissues

Expression patterns of *AaBAM* genes at different tissues (flower bud, root, twig, leaf and young fruit) were analyzed using RNA-Seq data. Flower bud, root, twig, leaf and young fruit were collected from *Annona atemoya*, and the tissues were stored at −80 °C for RNA extraction and transcriptome analysis. RNA was extracted using the Trizol method as described by Ma et al. [41]. The FPKM values were calculated by the Cufflinks pipeline (http://cufflinks.cbcb.umd.edu/ (accessed on 11 January 2023)), and the FPKM analysis results are shown in Appendix A. Genes with no expression (FPKM values equal “0” in all tissues) were filtered. The tissue differential expression of genes is shown in the heatmap.

### 4.6. Plant Materials and Treatments, RNA Extraction and Quantitative qRT-PCR Analysis

The fruit of *Annona atemoya* Hort (Africa pride) generated in September was used as the test material when the pericarp was between yellow and green, and the scale grooves between scales were expanded. After harvest, the fruits of similar size and color were selected for the experiment, and the different temperatures were set at 15 °C, 28 °C, 32 °C. A total of 15 fruits were deposited at each temperature, and samples were collected after treatment for 0 day, 2 days, 3 days, 4 days, 6 days, 8 days and 10 days, respectively. Overall, 3 samples from each treatment were collected and stored at −80 °C after rapid treatment with liquid nitrogen.

Treatment with ethephon and IAA was performed in the following way: The fruit was soaked with 2g/kg (ethephon/water) of ethephon aqueous solution, between 100mg/kg (ethephon/water) IAA [42], and distilled water (control) to remove hormones of the tissue; it was then placed at room temperature. Samples at 0 day, 6 h, 12 h, 24 h, 3 days and 5 days, respectively, were taken. A total of 3 samples from each treatment were collected—the sampling part was pulp—and it was quickly treated with liquid nitrogen and store at −80 °C for preparation.

RNA was extracted using the Trizol methods described by Ma et al. [41]. The purity and quantity of extracted RNA were identified by agarose gel electrophoresis and Nanodrop spectrophotometer. Synthesis of cDNA was performed using reverse transcription kit AT311. SYBR Green qPCR Master Mix was used for qRT-PCR analysis in Quant StudioTM 6 Flex System real-time fluorescence quantitative PCR system. Each reaction was performed in biological triplicates and the data from real-time PCR amplification were analyzed using the 2 −ΔΔCT method. Sequences of the primers used in this study re shown in detail in Appendix A.

### 4.7. Vector Construction Subcellular Localization Analysis of AaBAM Genes

Primers were designed according to the ORF of *AaBAM1.2*, *AaBAM3*, and *AaBAM9*, as shown in Appendix A. After cloning and sequencing validation, they were linked to a vector, and then transformed into *DH5α* after PCR amplification. The clones were screened and the plasmid was extracted to obtain the fusion expression vector of GFP and target gene. The constructed vector plasmid was transferred into *Agrobacterium* EHA105 by electrotransformation, and cultured at 30 °C for 2 days. The *Agrobacterium* tumefaciens was cultured in a YEB liquid medium, and the OD600 was adjusted to 0.6. The epidermis of tobacco was injected and cultured under dark/light for 2 days. The tobacco leaves injected with labeled *Agrobacterium* were taken and made into slides, and observed and photographed with Nikon C2-ER laser confocal microscope.

## Figures and Tables

**Figure 1 ijms-24-10516-f001:**
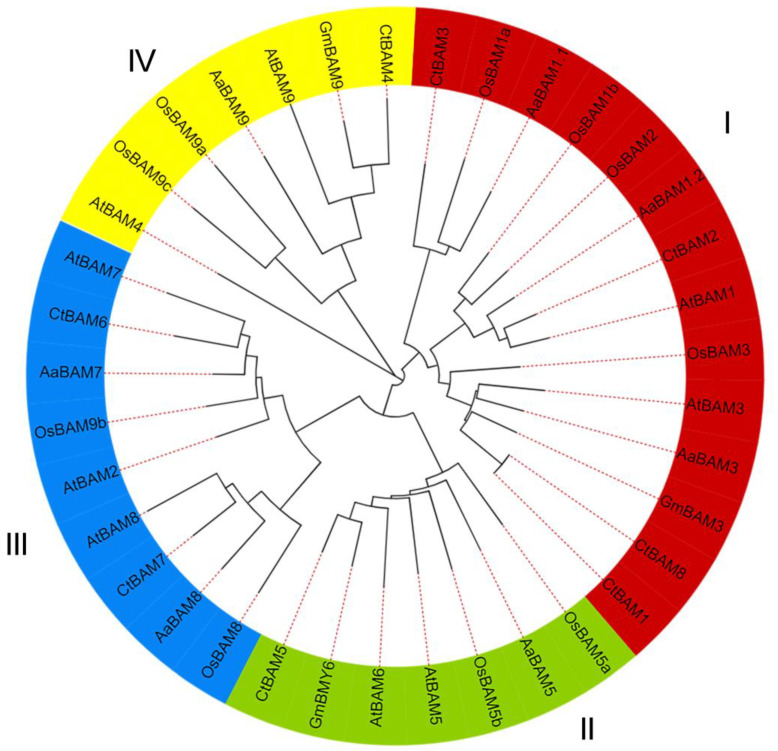
Unrooted phylogenetic tree representing relationships among BAM amino acids of *Arabidopsis*, *Glycine max*, *Citrus trifoliata*, and *Oryza sativa*. The different-colored arcs indicate different groups (or subgroups) of BAM genes. I represents the subfamily of Group I; II represents the subfamily of Group II; III represents the subfamily of Group III; IV represents the subfamily of Group IV.

**Figure 2 ijms-24-10516-f002:**
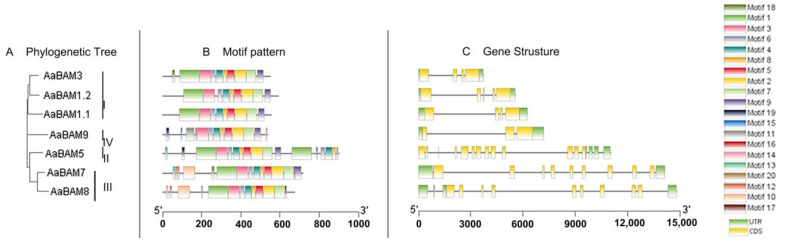
Phylogenetic relationships, gene structure and architecture of conserved protein motifs in *AaBAM* genes. (**A**) The phylogenetic tree was constructed based on the full-length sequences of Annona atemoya proteins using MEGA 7 software. (**B**) The motif composition of *AaBAM* proteins. The motifs, numbers 1–20, are displayed in different-colored boxes. The sequence information for each motif is provided in Appendix A. (**C**) Exon–intron structure of *AaBAM* genes. Yellow boxes indicate CDS; black lines indicate introns; green boxes indicate UTR. The length of protein and genes can be estimated using the scale at the bottom.

**Figure 3 ijms-24-10516-f003:**
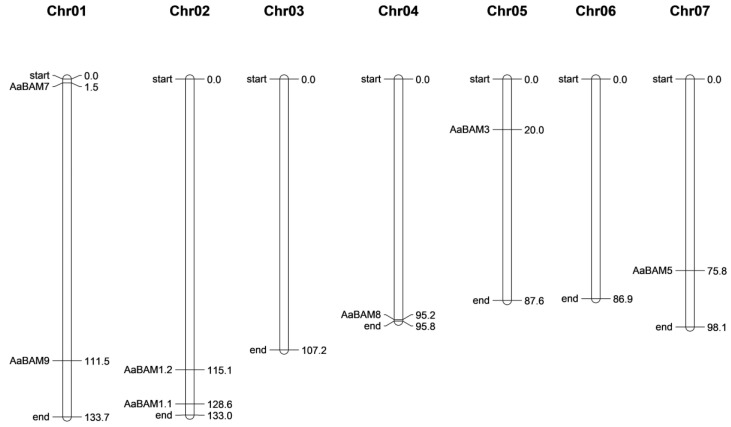
Distribution of *AaBAM* genes in chromosome (Chrs).

**Figure 4 ijms-24-10516-f004:**
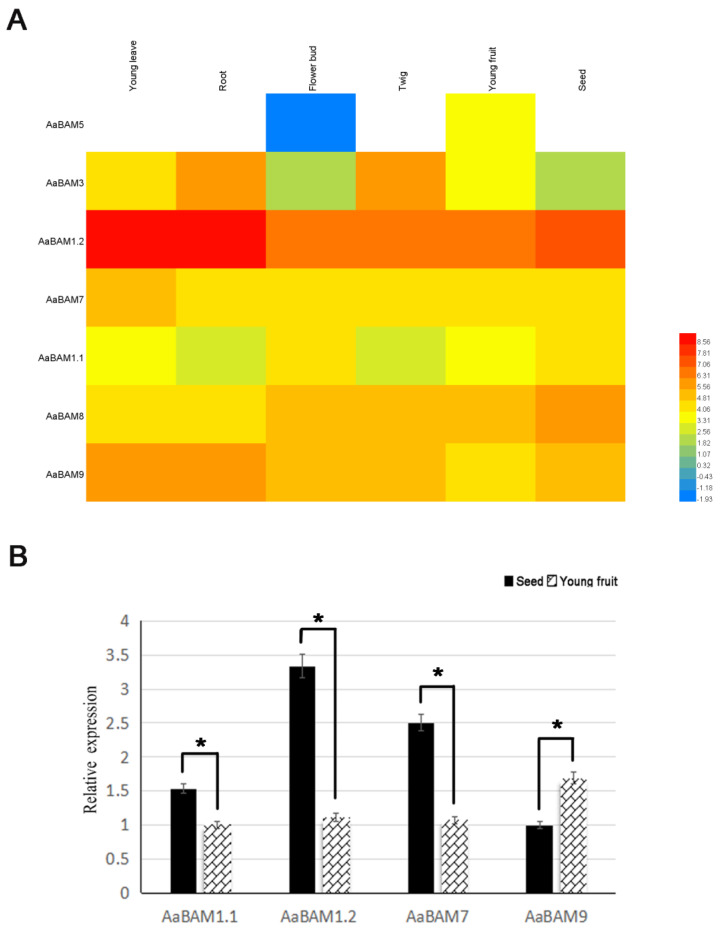
Expression profiles of the *AaBAM* genes. (**A**) Clustering of expression profiles of *AaBAM* genes in six samples including different tissues. (**B**) Expression analysis of 4 *AaBAM* genes in 2 representative samples by qRT-PCR. The expression value of *AaBAM1.1* in young fruit was taken as 1. ** p* ≤ 0.05.

**Figure 5 ijms-24-10516-f005:**
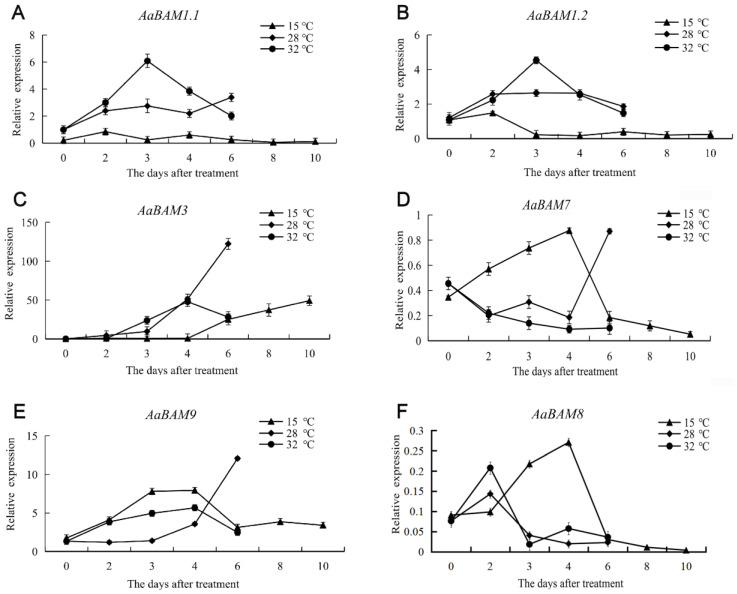
Expression analysis of 6 *AaBAM* genes at three representative temperature by qRT-PCR. Data were normalized to actin gene and vertical bars indicate standard deviation. The expression value of *AaBAM1.1* in 0 day fruits (not treated) was takeen as 1. (**A**) The expression of *AaBAM1.1* under temperature of 15 °C, 28 °C, and 32 °C. (**B**) The expression of *AaBAM1.2* under temperature of 15 °C, 28 °C, and 32 °C. (**C**) The expression of *AaBAM3* under temperature of 15 °C, 28 °C, and 32 °C. (**D**) The expression of *AaBAM7* under temperature of 15 °C, 28 °C, and 32 °C. (**E**) The expression of *AaBAM9* under temperature of 15 °C, 28 °C, and 32 °C. (**F**) The expression of *AaBAM8* under temperature of 15 °C, 28 °C, and 32 °C.

**Figure 6 ijms-24-10516-f006:**
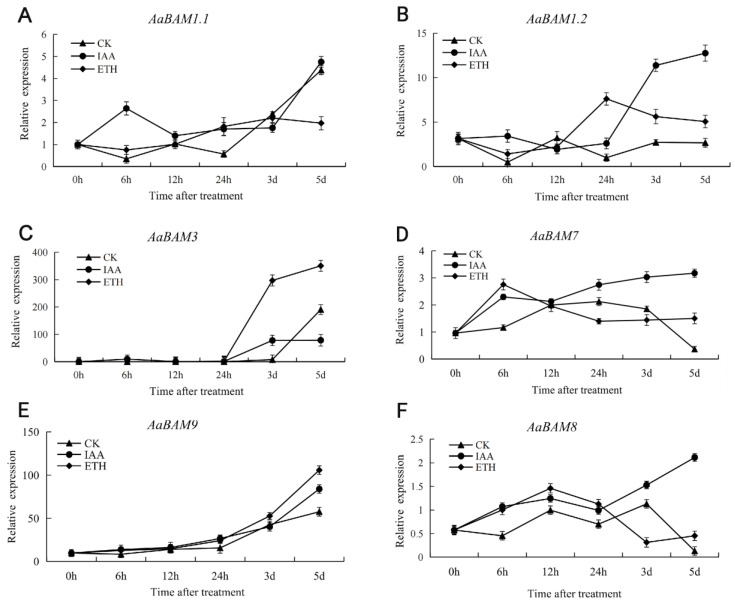
Expression analysis of 6 *AaBAM* genes in three representative phytohormone treamentby qRT-PCR. Data were normalized to actin gene and vertical bars indicate standard deviation. The expression value of *AaBAM1.1* in 0 h fruits (not treated) was taken as 1. CK, cytokinin; IAA, auxin; ETH, ethylene. (**A**) The expression of *AaBAM1.1* under phytohormone treatment of CK, IAA, ETH. (**B**) The expression of *AaBAM1.2* under phytohormone treatment of CK, IAA, ETH. (**C**) The expression of *AaBAM3* under phytohormone treatment of CK, IAA, ETH. (**D**) The expression of *AaBAM7* under phytohormone treatment of CK, IAA, ETH. (**E**) The expression of *AaBAM9* under phytohormone treatment of CK, IAA, ETH. (**F**) The expression of *AaBAM8* under phytohormone treatment of CK, IAA, ETH.

**Figure 7 ijms-24-10516-f007:**
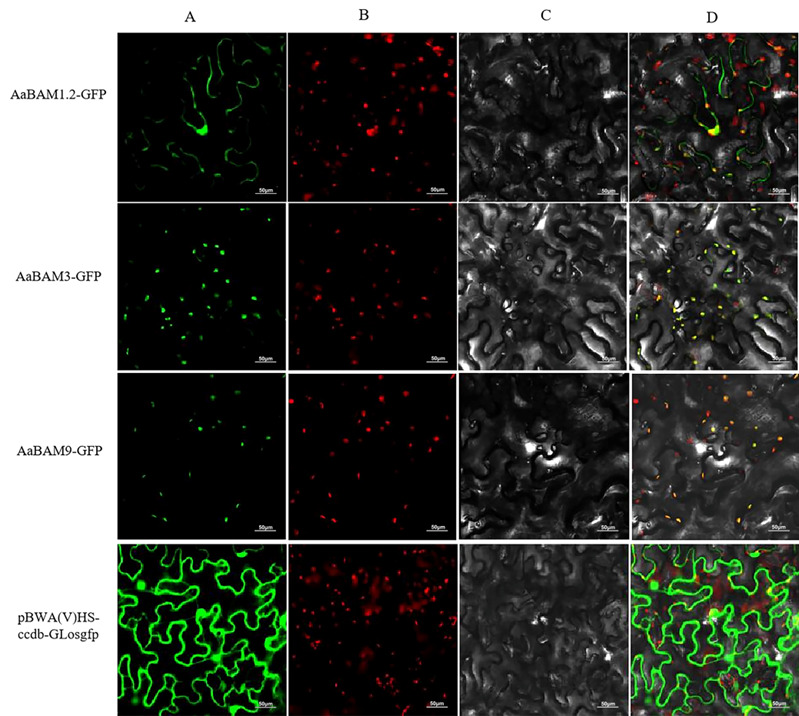
Subcellular localization of the full-length 3 AaBAM protein. (**A**) Fluorescence channel; (**B**) chloroplast channel; (**C**) bright channel; (**D**) merge channel. The constructed vector was transferred into tobacco leaves.

## Data Availability

The data presented in this study are available in Appendix A here.

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
