# Peer review of "Genome-Wide Investigation of BAM Gene Family in Annona atemoya: Evolution and Expression Network Profiles during Fruit Ripening"

_ijms, 2023, doi:10.3390/ijms241310516_

Round 1

Reviewer 1 Report

Manuscript entitled: Genome-wide investigation of BAM gene family in Annona atemoya: evolution and expression network profiles during fruit ripening" contains interesting information. Suggestions and comments on the manuscript are provided below.

- Introduction: there is no information on the studied species Annona atemoya, e.g. on the area of cultivation, place and harvest, taxonomic affiliation, economic importance tip.

- Line: 60-64 - this does not match Introduction. At this point, it should be written why this type of research was carried out, as well as the purpose and research hypothesis of the work

- Results: captions under all Figures do not comply with the requirements of the journal

- Throughout the manuscript, please pay attention to the Latin spelling of the species - it should be written in italics

- There are no Conclusions - please list the most important conclusions summarizing the research in a separate point

- Materials and Methods: no information on the statistical processing of test results.

- Line: 362 - Wrong spelling Supplemental Table - see instructions for authors

- Line: 371 - no information about Supplementary Materials entered - References: - incorrect spelling of cited journals - see requirements for authors

Author Response

Dear Reviewers:

Thanks for you letter for the comments concerning our manuscript entitled “ Genome-wide investigation of BAM gene family in Annona atemoya: evolution and expression network profiles during fruit ripening”. Those comments are valuable and helpful to revising and improving our paper. We have studied those comments carefully and have made correction which we hope to meet with approval. The main corrections and responds to the comments are as follow:

Introduction: there is no information on the studied species Annona atemoya, e.g. on the area of cultivation, place and harvest, taxonomic affiliation, economic importance tip.

Correction: We have added the background information of Annona atemoya, including taxonomy information, economic value, distribution area, fruit characteristics, industrial status, etc(Line: 53-59).

- Line: 60-64 - this does not match Introduction. At this point, it should be written why this type of research was carried out, as well as the purpose and research hypothesis of the work.

Correction: We have included in this section the purpose and significance of our research, which is based on the hypothesis that AaBAMs regulate starch degradation and thus affect fruit ripening. We aim to explore effect of β-amylase gene family on fruit ripening(Line: 67-69, 72-74).

- Results: captions under all Figures do not comply with the requirements of the journal.

Correction: it has been correction and comply with the requirements of the journal.

- Throughout the manuscript, please pay attention to the Latin spelling of the species - it should be written in italics.

Correction: We carefully checked and corrected with the writing issues.

- There are no Conclusions - please list the most important conclusions summarizing the research in a separate point.

Correction: We added a Conclusions in the last paragraph of the discussion part, summarizing what we consider are the most important conclusions(Line: 336-348).

- Materials and Methods: no information on the statistical processing of test results.

Correction: We carefully examined the materials and methods that were described incomplete, and added more information about RNA-seq and qRT-PCR(Line: 387--388, Line: 408-411).

- Line: 362 - Wrong spelling Supplemental Table - see instructions for authors.

Correction: it has been modified to comply with the requirements of the journal.

- Line: 371 - no information about Supplementary Materials entered - References: - incorrect spelling of cited journals - see requirements for authors.

Correction: Thank for the reviewer for their careful and meticulous works. The Supplementary Materials has been added, and the references have been revised according to the requirements of the journal.

We appreciate for you warm work earnest, and hope that the correction will meet with approval.

Once again ,thank you very much for you comments and suggestions!

Reviewer 2 Report

This study is aimed to investigate BAM gene family in Annona atemoya inclusing evolution and expression network profiles during fruit ripening. The study design is acceptable. The study presents some interesting and acceptable results. However the study needs some revisions before possiblle consideration for publication.

Suggestions:

L13 and L60:  β-amylase ... or  β-Amylase ...

L104: Letter type for figure 1 title is inconsistent. Please include I, II, III and IV into the title

L133: Space missing between 'genes' and 'Figure3'.

L153: Please indicate possible significant differences above columns.

L177: Why is the order 28, 32, 15 and not 15, 28, 32 in the figures?

L191: Please giv explanation for CK, IAA, and ETH in the title.

L286: A point is missing at the end of the sentence. A space is missin between 'signal' and '(Figure 6E)'

L305: 'hidden Markov model' or 'Hidden Markov Model'?

L309:  ... BAM domain. Annona atemoya ????

There is no Consclusion section. Please provide.

Formatting of References is inconsistent. Please consult with author guide.

Author Response

DearReviewers:

Thanks for you letter for the comments concerning our manuscript entitled “ Genome-wide investigation of BAM gene family in Annona atemoya: evolution and expression network profiles during fruit ripening”. Those comments are valuable and helpful to revising and improving our paper. We have studied those comments carefully and have made correction which we hope to meet with approval. The main corrections and responds to the comments are as follow:

L13 and L60:  β-amylase ... or  β-Amylase ...

Correction: The entire manuscript has been revised to be unified as β- amylase.

L104: Letter type for figure 1 title is inconsistent. Please include I, II, III and IV into the title.

Correction: I, II, III and IV have been included into the title.

L133: Space missing between 'genes' and 'Figure3'.

Correction: Space have been added between 'genes' and 'Figure3'.

L153: Please indicate possible significant differences above columns.

Correction: Significant differences have been added above columns in Figure 4.

L177: Why is the order 28, 32, 15 and not 15, 28, 32 in the figures?

Correction: The order of temperature has been changed to the uniform format of 15, 28, and 32.

L191: Please give explanation for CK, IAA, and ETH in the title.

Correction: The explanation for CK, IAA, and ETH have been added in the title.

L286: A point is missing at the end of the sentence. A space is miss in between 'signal' and '(Figure 6E)'.

Correction: The point and space have been added.

L305: 'hidden Markov model' or 'Hidden Markov Model'?

Correction: The entire manuscript has been revised to be unified as 'Hidden Markov Model'.

L309:  ... BAM domain. Annona atemoya ????

Correction: The “Annona atemoya” have been deleted.

There is no Consclusion section. Please provide.

Correction: Correction: We added a Conclusions part in the last paragraph of the discussion, summarizing what we consider are the most important conclusions (Line: 336-348).

Formatting of References is inconsistent. Please consult with author guide.

Correction: The references have been revised according to the requirements of the journal.

We appreciate for Reviewers’ warm work earnest, and hope that the correction will meet with approval.

Once again ,thank you very much for you comments and suggestions!